# Importance of Insoluble-Bound Phenolics to the Antioxidant Potential Is Dictated by Source Material

**DOI:** 10.3390/antiox12010203

**Published:** 2023-01-15

**Authors:** Fereidoon Shahidi, Abul Hossain

**Affiliations:** Department of Biochemistry, Memorial University of Newfoundland, St. John’s, NL A1C 5S7, Canada

**Keywords:** insoluble-bound phenolics, food matrix, distribution, antioxidant potential, bioactivity

## Abstract

Insoluble-bound phenolics (IBPs) are extensively found in the cell wall and distributed in various tissues/organs of plants, mainly cereals, legumes, and pulses. In particular, IBPs are mainly distributed in the protective tissues, such as seed coat, pericarp, and hull, and are also available in nutritional tissues, including germ, epicotyl, hypocotyl radicle, and endosperm, among others. IBPs account for 20–60% of the total phenolics in food matrices and can exceed 70% in leaves, flowers, peels, pulps, seeds, and other counterparts of fruits and vegetables, and up to 99% in cereal brans. These phenolics are mostly covalently bound to various macromolecules such as hemicellulose, cellulose, structural protein, arabinoxylan, and pectin, which can be extracted by acid, alkali, or enzymatic hydrolysis along with various thermal and non-thermal treatments. IBPs obtained from various sources exhibited a wide range of biological activities, including antioxidant, anti-inflammatory, antihypertensive, anticancer, anti-obesity, and anti-diabetic properties. In this contribution, the chemistry, distribution, biological activities, metabolism, and extraction methods of IBPs, and how they are affected by various treatments, are summarized. In particular, the effect of thermal and non-thermal processing on the release of IBPs and their antioxidant potential is discussed.

## 1. Introduction

Phenolic compounds are secondary metabolites distributed in various plants. Phenolic compounds in plants play a versatile role in regulating growth as a chemical messenger or an internal physiological regulator. They are responsible for protecting plants from ultraviolet (UV) radiation by absorbing harmful short high energy wavelengths, resulting in less oxidative stress. They are also involved in defense mechanisms against pathogen attacks and mechanical injury and play a key role in symbiosis and pollination [1,2]. Moreover, phenolic compounds influence the growth hormone auxin (IAA) and also play a key role in the color, flavor, bitterness, and astringency of foods. For example, phenolics, mainly tannins, are responsible for providing the bitter taste of unripe fruits and vegetables because of their interaction with salivary glycoproteins. Flavonoids such as apigenin, quercetin, and kaempferol can bind to a plasma membrane receptor, which interferes with the movement of polar auxin components and affect plant growth and architecture [3].

Over 8000 phenolic compounds have been identified from different sources, including fruits, vegetables, teas, spices, herbs, and whole grains, and they are mainly phenolic acids, flavonoids, stilbenes, and lignans. These are powerful natural antioxidants and have the potential to demonstrate inhibitory activity against chronic ailments such as cancer and cardiovascular diseases (CVDs) by binding to plasma membrane receptors [4,5]. In particular, phenolic compounds have the potential to inhibit LDL cholesterol and DNA oxidation as well as α-glucosidase, tyrosinase, and the formation of advanced glycation end-product (AGEs) [6]. Furthermore, phenolic compounds are becoming very popular in several industries, including the food, nutraceutical, pharmaceutical, cosmetic, packaging, and textile industries, due to their antioxidant, antimicrobial, and coloring properties [7,8]. Apart from their structural variations, phenolics exist in various forms, such as soluble-free, soluble-bound, including esterified, etherified, or glycosylated, and insoluble-bound forms, based on their association with food matrices in plants [2,4]. Usually, free phenolics do not form a chemical bond with other molecules and exist in free forms within the plant cell vacuole, while esterified phenolics are covalently bound to sugars or other molecules such as fatty acids and glucuronic acid through an ester bond. In contrast, insoluble-bound phenolics (IBPs) are covalently bound with structural molecules such as hemicellulose, cellulose, pectin, and structural protein, mainly rod-shaped (Figure 1) [4]. The phenolic profile of IBPs is less diverse than the free phenolic fraction and is mainly phenolic acids, condensed tannins, and other low-molecular-weight phenolics, which are abundant in cereals, mostly phenolic acids. For example, ferulic and *p*-coumaric acids in whole grain oats are present at 10 to 100 times higher concentrations when compared to the free phenolic fraction [9]. The content of IBPs is mainly dependent on the source materials, processing methods, and extraction techniques. For example, up to 90% of the total phenolic acids in rice, mainly bran, are in the bound form, while around 50–95% of the phenolic compounds that exist in fruits are in the bound form [10]. Generally, fruits and vegetables have a higher content of soluble free or soluble conjugated phenolics and a lower level of IBPs, around 25% of the total phenolic content (TPC). However, the opposite scenario is observed in terms of IBPs of various grains. IBPs are not absorbed in the small intestine but absorbed in the human gastrointestinal tract via colonic fermentation, followed by the release of IBPs, improving bioaccessibility and bioavailability. Nevertheless, IBPs are often ignored as they require special treatment to release them from the food matrix, and hence, their actual phenolic content and activities are underestimated. Therefore, IBPs are also known as “unextractable phenolics” or “non-extractable phenolics” due to the fact that these remain in the extraction residue and their covalent bond with macromolecules is hard to break with solvents, though they can be released upon acid, alkali, or enzymatic hydrolysis. In addition, processing methods also play a vital role in changing the content as well as the activities of IBPs. For instance, thermal processing, mainly cooking, of hawthorn reduced the content of total soluble phenolics but increased the IBP content [11]. However, the changes in different phenolic fractions and their impact on antioxidant potential upon processing are poorly understood. Therefore, this review summarizes the IBPs in different food matrices and their interactions with other molecules. Moreover, the effect of different processing methods on the IBPs, and their chemistry, metabolism, health benefits, and extraction methods are also discussed.

## 2. Synthesis and Transport of Phenolics in Plants

In plant cells, phenolics are mainly synthesized in the cytoplasmic surface of the endoplasmic reticulum and continue to the outer nucleus of cells. These synthesized phenolic compounds in the intracellular components are released and transported to the cell wall substances via the vesicle transfer system and form the complex of IBPs. These migrated phenolic compounds are bound to cell wall matrices, such as protein, cellulose, and pectin, among others, via covalent bonding, which play a major role in building cell wall materials [2,12]. However, the mechanism of transformation and formation of IBPs is not yet fully understood. The synthesized phenolics could be transported into cell wall substances as IBPs or vacuoles as soluble free phenolics through vesicle transfer membrane-mediated transport systems. The migration of IBPs includes the Golgi body and cytoplasmic vesicles, which are lipid bilayer systems, facilitating the transformation of phenolic compounds into cell wall components [4]. Moreover, phenolics in the cytoplasmic vesicles can reach the cell wall matrix via ABC transporters [2]. Various types of enzymes, namely phenylalanine ammonia lyase (PAL), erythrose-4-phosphate, cinnamate-4-hydrolxylase, *o*-methyl transferase, and *p*-coumarate-3-hydroxylase, are responsible for forming the structure of phenolic compounds, resulting in several classes of phenolics, including phenolic acids, flavonoids, and other types of phenolics [6]. Phenolics in plants are mainly derived from phenylalanine, and in some cases, tyrosine. The formation of *trans*-cinnamic acid from phenylalanine is catalyzed by PAL, whereas *p*-hydroxycinnamic acid from tyrosine is catalyzed by tyrosine ammonia-lyase (TAL) [13]. This *trans*-cinnamic acid is further modified into *p*-coumaric acid and then caffeic, ferulic, and sinapic acids by cinnamate-4-hydrolxylase, *p*-coumarate-3-hydroxylase, *o*-methyl transferase, and hydroxylase and *o*-methyl transferase (Figure 2).

Similarly, *p*-coumaric acid is produced by TAL, and then it is transformed into different types of phenolic acids. On the other hand, flavonoids are synthesized through a combination of *p*-coumaroyl CoA and three molecules of malonyl CoA. This process is mainly initiated by the action of chalcone synthase (CHS) and produces naringenin chalcone, which later converts into flavonone, a process catalysed by chalcone isomerase. This flavonone can be converted into various flavonoids such as flavononol, flavonol, flavone, isoflavone, catechins, and anthocyanins via enzymatic action [6].

## 3. Classification Based on the Structure

Phenolics can be categorized into different groups: phenolic acids, flavonoids, tannins, stilbenes, lignans, and coumarins. Phenolic acids are a major type of phenolic compound and usually occur in various conjugated forms rather than the free type. Phenolic acids comprise a phenyl group substituted by one carboxylic group with one or more hydroxyl groups. Phenolic acids consist of hydroxycinnamic acids (C6–C3), mainly caffeic, ferulic, *p*-coumaric, and sinapic acids. Meanwhile, the loss of a C-2 moiety leads to the formation of hydroxybenzoic acids (C6–C1), mainly vanillic, syringic, ellagic, *p*-hydroxybenzoic, and protocatechuic acids (Figure 3). Based on the substitution and functional groups, such as methoxy and hydroxyl groups, the differentiation occurs among the individual phenolic acids. Flavonoids are a group of more than 4000 phenolic compounds and are composed of two aromatic rings (A and B) linked by a third ring (C) in the C6–C3–C6 form. According to the different substitution patterns, such as hydroxyl and methoxy groups, flavonoids can be classified into different sub-categories, mainly flavones, flavanones, flavanonols, flavonols, flavanols, isoflavones, and anthocyanidins (Figure 3).

The commonly encountered flavonoids which are widely distributed in fruits and vegetables are catechin, quercetin, naringenin, daidzein, and cyanidin glycoside. However, anthocyanins play an important role in providing color in plants, especially fruits and vegetables, and these colors can change depending on the pH, being yellow to red in acidic, blue to violet in alkaline, and green to purple in neutral solutions. Moreover, depending on the chemical structure, tannins can be classified into two classes of hydrolyzable (e.g., ellagitannins) and condensed (e.g., proanthocyanidins). Hydrolyzable tannins contain a central glucose core, esterified to gallic or ellagic acid, while condensed tannins are oligomers and polymers of flavonoids. Furthermore, stilbenes possess a carbon skeleton of C6–C2–C6, which includes resveratrol. Based on the level of polymerization, stilbenes can be categorized into several groups, mainly monomers, dimers, trimers, tetramers, and hexamers. In addition, coumarins are referred to as benzopyrone with the basic skeleton of C6–C3. Lignins are the polymers of monolignols, and their structure is constructed from two phenylpropanoid units (C6–C3–C3–C6) [6,13,14].

## 4. Classification and Localization of Phenolics Based on Their Association with Food Matrix

Based on the distribution in nature and location in plants, phenolic compounds either occur in free or bound (soluble and insoluble) forms. Generally, free phenolics are distributed within the plant cell vacuole, do not bind physically and chemically with other molecules, and can be extracted using organic or polar solvents. Soluble-bound, including soluble-glycosylated and soluble-esterified-bound, are covalently bound or esterified to sugars and low-molecular-weight macromolecules. Soluble-bound phenolics are attached to one or more sugar units via carbon–carbon linkages (C-glycosides) or a hydroxyl group (O-glycosides). For instance, hydroxyl groups of cell wall substances can form ester bonds with the carboxyl group of phenolic acids (e.g., cinnamic and benzoic acids) and can also create ether bonds with the hydroxyl groups of phenolic compounds. In contrast, most IBPs are bound to cell wall substances such as cellulose, hemicellulose, pectin, arabinoxylan, lignin, and structural proteins through ether, ester, and C–C bonds. Moreover, some phenolic compounds in the IBP fraction might be bound with the food matrix via hydrophobic interactions (mainly with structural proteins) or hydrogen bonds (Figure 4) [2,4,10].

Generally, IBPs include high-molecular-weight compounds (e.g., condensed tannins) and soluble phenolic compounds consisting of low- and medium-molecular-weight compounds (e.g., simple phenols, tannins, and flavonoids). IBPs are present in comparatively high amounts (20–60%) in plants compared to the soluble phenolics and are not extracted using an extraction medium since they are bound covalently to the insoluble macromolecules [2,15]. For instance, Burlini and Sacchetti [16] summarized the bound phenolics in various foods and reported that the apple contained 6.5% of IBPs, while this was 88% for brown rice.

Bound phenolics in plants are mainly distributed in the protective tissues, such as the seed coat, pericarp, and hull, and are also available in nutritional tissues, including germ, epicotyl, hypocotyl radicle, and endosperm, among others. This is because phenolics in protective tissues play an important role in protecting tissues from harmful organisms such as insects, pathogens, and herbivores, as well as UV light and high temperatures. Moreover, the major nutritional components (e.g., carbohydrates, proteins, and lipids) are mainly distributed in nutritional tissues rather than protective tissues, resulting in high soluble and bound phenolics in the seed coat, pericarp, and hull, including the epidermis, chlorenchyma, hypodermis, parenchyma, palisade, and endothelium cells, among others [2,10]. For example, corn (baby corn, popcorn, and sweet corn) pericarps were found to contain 74–83% of bound phenolics, while free phenolics were abundant in nutritive tissues (germ and endosperm) [17]. Not only the content but also the types of individual phenolic compounds are higher in the protective tissues when compared to nutritional tissues. For instance, a higher number of bound phenolic compounds (syringic, vanillic, caffeic, ferulic, *p*-coumaric, and isoferulic acids, and kaempferol and quercetin) were identified in the pericarp of sweetcorn, while baby corn endosperm contained only syringic acid in the bound phenolic fraction [17]. In addition, the major parts of leaves (e.g., epidermal cells, subsidiary cells, guard cells, and epidermal hairs/trichomes) and stems (e.g., epidermal cells, chlorenchyma cells, parenchyma cells, and collenchyma cells) are a good source of phenolic compounds [18,19]. However, the content and type of individual phenolic compounds varies among different species or even in the same species. This could be due to the different geographical locations, climatic conditions, levels of stress, harvest times, and environmental factors as well as extraction methods. For example, different varieties of barley (blue, black, and yellow) contain bound phenolics between 54.6 and 88.9%, while brown rice contains about 88% of bound phenolics [20].

## 5. Interaction of Phenolics with Other Compounds

Normally, proteins, polysaccharides, phenolics, and other compounds do not interact with each other and are separated into various cell compartments. However, external stimuli trigger the intracellular contact, resulting in numerous subsequent reactions (oxidation, adsorption, migration, and solubilization) which forces them to interact with each other [21]. For example, phenolic compounds, mostly simple phenolic acids and phenylpropanoids, can form complexes with carbohydrates via hydrophobic interactions and hydrogen bonding. Moreover, phenolic acids can be bound to oligosaccharides, mainly cyclodextrin, through non-covalent complexation [9,21]. On the other hand, the complexation between phenolic acids and proteins is dependent on the affinity of the phenolic acid for water, molecular weight, number and position of hydroxyl groups, pH condition (neutral and basic), and structural flexibility [22]. For instance, due to hydrophobic interactions between the hydrophobic region of the protein and the aromatic ring of tannins, they can interact with each other under favorable conditions, depending mainly on the length, size, and pH. The thermal stability of proteins can improve upon interaction with phenolic compounds [9]. However, phenolic–protein interactions could affect the function of proteins, phenolics, and their antioxidant activities, as well as their bioavailability. For example, the radical scavenging activity of the flavonoid, epigallocatechin gallate (EGCG), was found to be lower in the presence of milk protein [15]. In addition, phenolics can interact with dietary fiber via covalent and non-covalent interactions. Phenolic–dietary fiber interaction is mainly dependent on the molecular weight, degree of hydroxylation, methoxylation, methylation, esterification, hydrogenation, and glycosylation of phenolic compounds. For example, the interaction between phenolics and oat β-glucans was investigated, and it was found that the presence of four or more hydroxyl groups reduced the binding interactions, while the opposite scenario was seen for those that had three or fewer hydroxyl groups [23]. The same study also suggested that the adsorption capacity of flavonoids into oat β-glucan was in the order of flavonol > flavone > flavanone > isoflavone.

## 6. Effect of Processing on the Release of Insoluble-Bound Phenolics

Various processing methods, including thermal and non-thermal processing (high pressure, fermentation, and germination), have been used to release IBPs from different sources, and most of them positively affect the content and their antioxidant activities due to the release of IBPs (Table 1).

### 6.1. Non-Thermal Processing

High-pressure processing (HPP) is a non-thermal operation, which is useful to preserve foods containing heat-sensitive components due to its higher extraction yields, minimum thermal degradation, and shorter time. HPP helps to bring faster diffusion and cell disruption, improving solvent accessibility and leading to better extraction [26]. For example, Zhou et al. [24] applied a non-thermal treatment, ultra-high pressure (UHP), to enhance the content of free, esterified, and IBP fractions in oil palm (*Elaeis guineensis* Jacq.) fruits, thus leading to a significantly increased TPC and total flavonoid content (TFC), especially those of the insoluble-bound phenolic fraction. Moreover, the antioxidant activity and the content of individual phenolic compounds, mainly caffeic acid, increased in all three fractions by about 30% upon UHP. This could be due to the ability of UHP to destroy the cell walls of oil palm fruits, resulting in their enhanced bioaccessibility [24]. Similarly, free, esterified, and IBPs were extracted using UHP from mango leaves, and results suggested that UHP significantly influenced TPC, TFC, the contents of individual compounds, and antioxidative and cytoprotective properties, mainly those related to IBPs [25]. The positive effect of UHP on the yield of phenolics could be linked to the disruption of the cell wall or the chemical bonds between the cellular components, such as proteins, cellulose, hemicellulose, lignin, and phytochemicals [25]. On the other hand, the free, esterified, and IBPs of the sea cucumber (*C. frondosa*) body wall was investigated using HPP pre-treatment (200, 400, and 600 MPa for 5, 10, and 15 min) [26]. Treatment of 600 MPa for 10 min improved the TPC, TFC, antioxidant activity, and the contents and numbers of phenolic compounds. HPP could enhance solvent penetration into the sea cucumber body wall through the interference of the cellular matrices, which may improve mass transfer and permeability, causing a better release of phenolics, including IBPs. Likewise, the contents and number of individual phenolic compounds and their bioactivities of sea cucumber processing discards increased upon HPP [27].

Fermentation is a traditional non-thermal food processing method where sugar molecules are converted to lactic acid, ethanol, and gas via microbial action. Microorganisms secrete a variety of extracellular enzymes, including proteases, carbohydrases, and lipases, during fermentation to break down macromolecules such as starch, cellulose, proteins, and phenolic polymers into smaller components (e.g., glucose, peptides, free amino acids, and phenolic derivatives). Moreover, fermentation could release IBPs from the cell wall substances through the degradation of cell walls by cellulase, hemicellulase, esterase, amylase, pectinase, and glucanase [2]. Fermentation improves the antioxidant activity of fermented foods, which could be related to the liberation of IBPs by cell-wall-disintegrating enzymes. For instance, Shumoy et al. [29] determined the soluble and bound phenolics from a traditional fermented pancake (Injera) and found that fermentation increased the contents of soluble phenolics by 92-150% after 72 h and bound phenolics by 13–55%, as fermentation progressed from 0 to 120 h. The improvement of bound phenolics could be related to the break down of ester linkages via enzymes such as xylanases, esterases, and phenol oxidases. However, the percentage of bound phenolic improvement was lower compared to the soluble fraction, which could be linked to the conversion of soluble phenolics from bound phenolics upon fermentation. Furthermore, the organic acids produced during the lactic acid bacteria (LAB) fermentation could hydrolyze bound phenolics from the cellular substances, leading to the release of bound phenolic compounds [29]. In contrast, Yeo et al. [30] suggested that the fermentation of lentil hull significantly decreased the content of IBPs, indicating their liberation upon fermentation. However, the efficiency of bioconversion from IBPs to soluble phenolics was low, suggesting the loss of the released bound phenolics during fermentation. Moreover, all IBPs were not converted into bioavailable soluble phenolics, and this could be due to their structural variations during fermentation.

On the other hand, germination stimulates the rupturing of the dormancy of seeds by sprouting and growth, activating cell metabolism. Therefore, structural macromolecules such as starch and proteins can be converted into smaller molecules by hydrolytic enzymes released from activated cells, affecting the content of bound phenolics and their formation [2]. The effect of germination on the free phenolics and IBPs of mustard grains (*Brassica nigraand* and *Sinapsis alba*) was investigated and found to positively affect the content as well as the antioxidant activity of *S. alba* [31]. However, the opposite scenario was found for the *B. nigra*, suggesting the conversion of IBPs to soluble phenolics. The liberation of IBPs could be associated with the increased total volume of the cell wall associated with cell division (biosynthesis) during germination [2]. On the other hand, the ratio of IBPs to soluble phenolics of lentils was investigated to monitor changes in antioxidant activity upon germination [32]. Results indicated that the overall ratio of IBPs to soluble phenolics improved during germination, and this could possibly be due to the conversion of phenolics from soluble into insoluble-bound form. The decrease in soluble phenolics could be related to the transportation from the intracellular space to cell walls or the degradation of flavonoids by reactive oxygen species (ROS) [32].

### 6.2. Thermal Processing

Thermal treatments such as roasting, extrusion cooking, boiling, hot drying, steam explosion, and infrared and microwave heating not only help to improve the flavor, texture, and taste of foods but also release biomolecules. For example, Li et al. [11] stated that thermal (dried, lightly cooked, and well-cooked) processing of hawthorn significantly increased IBPs, but decreased soluble phenolics. Boiling can disrupt the covalent and hydrogen bonds between phenolics and cellular components, releasing more IBPs, while soluble phenolics could degrade under heat treatment or convert into IBPs upon condensation reactions with sugars and proteins via hydrogen bonds [11]. In contrast, hydrothermal (boiling) processing of lentils increased the content of soluble phenolics and decreased IBPs, suggesting their possible release from cellular components [12]. However, the reduction in IBPs was around four times higher than the increase in the content of soluble phenolics. This could be due to the conversion of IBPs into soluble phenolics and/or the loss of bound phenolics upon heat treatment. The loss of bound phenolics could be linked to the formation of irreversible covalent bonds with other macromolecules, including starch, cellulose, and proteins, that cannot be liberated via regular IBP extraction procedures [12]. Similarly, thermal pre-treatment decreased the overall bound phenolics of virgin *Camellia oleifera* seed oil and increased free phenolics [33]. On the other hand, Peng et al. [34] applied microwave and enzymatic treatments and their combination to release IBPs from grapefruit peel and found that the combination of these treatments afforded the highest content of IBPs. Moreover, the combination of these treatments resulted in a weakening of the dietary fiber, which was confirmed using scanning electron microscopy (SEM), and in removing lignin, which was checked via X-ray diffraction and FT-IR.

## 7. Insoluble-Bound Phenolics in Various Food Matrices

### 7.1. IBPs in Fruits, Vegetables, Herbs, and Their Different Parts

IBPs are extensively distributed in various food matrices such as fruits, vegetables, legumes, pulses, cereal grains, teas, coffees, and various seeds and oilseeds (Table 2 and Table 3). The major phenolics in fruits and vegetables occur in the soluble free form, whereas IBPs account for 20–60% of the TPC in foods. However, leaves, flowers, peels, pulps, seeds, and other counterparts of fruits and vegetables are a rich source of IBPs and can reach ≥70% of TPC [10] (Table 2).

For example, Zhou et al. [24] determined the phenolic content of oil palm (*Elaeis guineensis* Jacq.) fruits using UHP and found that IBPs possessed the highest TPC (461.38 mg GAE/g), with a dominance of caffeic acid (11269.66 µg/g). Interestingly, the TPC of IBPs was higher (53.47%) than the sum of TPCs in the free and esterified phenolic fractions (46.55%). On the other hand, phenolics in *Rhus chinensis* Mill. fruits mainly existed in the esterified form, where IBPs had an almost four times lower TPC compared to the free phenolic fraction [68]. Moreover, the antioxidant activity of phenolics, mainly flavonoids, obtained from hawthorn berry fruit (*Crataegus pinnatifida*) was higher (35.3–37.8%) in the free phenolic fraction, followed by IBPs (25.0–27.0%). Epicatechin and protocatechuic acid were the main phenolics in IBPs [65]. Pico et al. [74] investigated the bound phenolics of six northern highbush blueberries and found that syringetin-3-*O*-glucoside was the most abundant bound phenolic compound, which was absent in the free phenolic fraction. Furthermore, free and bound phenolics were extracted from 19 potato genotypes, among them Longshu 7 contained the highest bound phenolics with a dominance of benzoic and caftaric acids [69]. Xue et al. [75] examined the release of bound phenolics from the insoluble dietary fiber of navel orange peel and determined their characteristics and mechanisms using mixed solid-state fermentation with *Aspergillus niger* and *Trichoderma reesei*. Scanning electron microscopy (SEM) indicated that the cell wall and crystalline structure of the peel were significantly decomposed upon fermentation. The major compounds released by fermentation were *p*-coumaric acid (1885.16 µg/g), quercetin (78.17 µg/g), naringin (18.21 µg/g), and vanillic acid (17.4 µg/g), which significantly enhanced the in vitro antioxidant potential. Similarly, the combination of microwave and enzymatic (ME) treatment was able to release IBPs from grapefruit peel and yielded the highest TPC (1.48 mg GAE/g) when compared to their individual treatment [34]. The SEM, X-ray diffraction, and FT-IR analyses confirmed that the structure of grapefruit peel insoluble dietary fiber was loosened upon ME, mainly with the removal of lignin. The major phenolic compounds were gallic (42.50 μg/g), ferulic (18.46 μg/g), and protocatechuic (6.16 μg/g) acids, which demonstrated strong antioxidant activity. In addition, phenolic compounds of brocade orange (*Citrus sinensis* L. Osbeck) peel existed as an esterified phenolic fraction followed by glycosylated, free, and IBPs. IBPs and esterified phenolic fractions contained ferulic and *p*-coumaric acids in abundance, but free phenolics contained mostly sinensetin, hesperidin, and nobiletin [76]. The presence of IBPs in fruit peels could suggest their role in a defense mechanism against adverse biotic (e.g., insect, pathogen, and herbivore attack) and abiotic (e.g., temperature and UV radiation) conditions.

Fruit pulp phenolics are mainly dominated by IBPs as they are attached to the cell wall matrix. For instance, the percentage of IBPs in Kainth (*Pyrus pashia*) fruit pulp is 68.24, which is around four times higher than the free phenolic fraction [56]. A total of 18 individual phenolic compounds were identified and quantified from the IBP fraction of Kainth fruit, where catechin was the main compound. Moreover, Xu et al. [66] examined the complex between the insoluble dietary fibers (IDF) and IBPs of lychee pulp using SEM and found that the complex became loose after alkaline hydrolysis. Ferulic acid was released from IDF upon hydrolysis, which was characterized using a confocal laser scanning microscope, and a C-O bond disruption was also confirmed using FTIR spectroscopy. Like fruit pulps, seeds are also a rich source of IBPs. For example, the percentages of IBPs in blackberry, black raspberry, and blueberry were 58.3, 63.01, and 57.06, respectively [53]. The order of the phenolic fraction was insoluble-bound > esterified > free, where quercetin 3-*O*-glucoronide, quercetin, epicatechin, *p*-coumaric acid, and gallic acid were dominant in the IBP fraction. Likewise, the percentage of IBPs in raspberry seeds was 28.77, where the major phenolic compounds were phenolic acids, mainly gallic, ellagic, *p*-coumaric, protocatechuic, and caffeic acids [54]. In addition, IBPs in araticum fruit accounted for 44.12, 20.74, and 19.94% of the TPC of pulp, seed, and peel, respectively [57]. Hence, the contents of IBPs remaining in fruits after juice extraction are not negligible, mainly in pulps, which contain flavonoids including catechin and epicatechin in abundance. Ambigaipalan et al. [67] investigated the soluble- and insoluble-bound phenolics of pomegranate by-products (mesocarp, outer skin, and divider membrane). Among them, soluble phenolic content (free and esterified) was higher than that of IBPs in all by-products, especially in the divider membrane (~44%). Similarly, phenolic compounds of industrial food wastes, including apple pomace, apple peel, pomegranate seed, pomegranate peel (PL), black carrot pomace, and chestnut shell (CS), were investigated [77]. The highest TPC and antioxidant activity were found in the soluble phenolic fraction of PL due to the presence of punicalagin derivatives. However, other wastes, including CS, had more phenolics (~45%) in the insoluble-bound form and showed strong antioxidant activity. Moreover, the IBPs in raspberry pomace without seeds and raspberry seeds were 26.47 and 28.78%, respectively, and they were dominated by phenolic acids such as gallic, ellagic, ferulic, *p*-coumaric, and protocatechuic acids [54].

Fruit leaves contain numerous antioxidants, including phenolics, mainly phenolic acids, flavonoids, tannins, and lignin. In particular, leaves are a rich source of IBPs that protect cells from oxidative damage. For instance, Chen et al. [51] analyzed the IBPs of 14 subtropical fruit leaves collected from the south of China and found that the contents of TPC released from the insoluble-bound fraction of *Musa sapientum*, *Artocarpus heterophyllus, Musa nana, Averrhoa carambola*, *Clausena lansium, Musa basjoo, Amygdalus persica*, and *Psidium guajava* leaves were 76.51, 72.58, 61.20, 53.77, 48.25, 46.80, and 42.13%, respectively. Furthermore, the IBP fraction contained catechin, rutin, isoquercitrin, and quercetin in abundance. Likewise, perilla leaves are also a good source of IBPs, mainly caffeic acid and scutellarein 7-*O*-diglucuronide. Nevertheless, red perilla leaves contained a higher percentage of IBPs (49.04%) when compared to green leaves (11.28%), which could be related to their distinct biosynthesis [50]. Zhang et al. [22] claimed that the TPC of the IBP fraction was even close to the sum of the TPC in the free and esterified phenolic fractions of mango leaves. Specifically, gallic acid was dominant in the IBPs, showing potential antioxidant and cytoprotective activities. On the other hand, phenolics, mostly flavonoids and tannins, in persimmon leaves mainly exist as free form, and their content varies between different geographical locations, cultivars, harvesting times, and drying methods [78,79]. In particular, persimmon leaves contained a higher phenolic content during the flowering stage when compared to the fruiting stage [80]. Yu et al. [48] determined the free, esterified, and IBPs in three different parts, including the leaf, flower, and stem, of *Lonicera japonica* and *L. macranthoides*. Free phenolics were the highest in terms of TPC and TFC, while caffeic acid, luteoloside, and isoquercitrin were abundant in the IBPs of leaves. Similar to this study, Xiang et al. [49] also found that the phenolic compounds of floral organs of two *Camellia* species flowers were mainly occurred as a free form, which was mainly flavonols, ellagitannins, procyanidins, phenolic acids, and flavanone. In particular, petals contained a higher content of phenolics, including isoquercitrin, astragalin, kaempferol-3-*O*-rutinoside, quercitrin, and afzelin, than the stamens. In addition, free, esterified, etherified, and IBPs of the fruit, leaves, stem, and root of *Terminalia sericea* were investigated, and the highest percentage (40.70–43.44) of IBPs was observed in the stem and root, which showed strong antioxidant activity [52].

### 7.2. IBPs in Cereals, Legumes, Pulses, and Other Seeds

The content of IBPs in cereals, such as corn (85–98.88%), barley (62–88%), rice (52–91%), oat (75–88%), wheat (75–83.18%), millet (22.23–73.63%), red sorghum (85.48%), and quinoa seeds (80%), varies among cultivars, body parts, seasons, and geographic locations, among others. For example, Bueno-Herrera et al. [60] reported that the IBPs in purple wheat (*Triticum aestivum* L.) contributed to 59 and 63% of the TPC in bran and flour, respectively. Among them, hydroxycinnamic acid and mainly *trans*- and *cis*-ferulic acids (over 70%), were abundant in coarse (95%) and fine (91%) brans. Similarly, the contribution of IBPs to the TPC of hard and soft wheat brans was significantly higher (~84%) than that of free and esterified fractions [37]. The increased concentration of IBPs at the outer layers of wheat could be linked to providing a physical barrier against insect and fungal pathogens. Moreover, Zhu et al. [35] analyzed the phenolics of six buckwheat varieties and found that the free phenolic fraction had higher quantities of phenolics, mainly quercetin, rutin, and kaempferol-3-*O*-rutinoside, than the bound form. However, the contents of gallic acid, *p*-hydroxybenzoic acid, 5-caffeoylquinic acid, and dihydromyricetin were higher in IBPs than the free phenolic fraction in most buckwheat varieties. Likewise, the highest bound phenolic content was observed in the middling and bran flour of buckwheat, where catechin and epicatechin were dominant [36]. On the other hand, ferulic and *p*-coumaric acids are the major IBPs in millets, which demonstrated a 38–99% ROS scavenging activity [38,39]. In particular, the IBPs of finger millet (*Eleusine coracana*) mainly contained protocatechuic acid (37.64%), vanillic acid (11.80%), ferulic acid (20.33%), and epigallocatechin gallate (9.62%) [81].

The contribution of IBPs to the TPC of barley was significantly higher than the soluble phenolics [40,41,42]. Specifically, the ratio of soluble to IBPs of barley ranged from 1:27 to 1:35 [40]. The major IBPs in barley varieties were gallic acid, benzoic acid, syringic acid, naringenin, and hesperidin, which showed a higher DPPH radical scavenging activity than the free phenolic fraction [42]. Moreover, Deng et al. [61] examined the phenolics of fourteen varieties of hulless barley; a higher phenolic content and antioxidant capacity were observed in IBPs than that of free and esterified phenolic fractions. Ferulic acid (266.9–744.2 μg/g), sinapic acid (1.35–15.72 μg/g), chrysoeriol-7-*O*-glucuronide (10.86–64.93 μg/g), and luteolin (3.01–12.56 μg/g) were the principal IBPs in most of the varieties. Likewise, ferulic acid, *p*-coumaric acid, and 8-5′ dehydrodiferulic acid dimers were discovered as the most common IBPs in brown rice, rice bran, and polished rice [82]. Furthermore, the content of 9-methyl-(8–8′)-cyclic dehydrodiferulic acid was almost 156 times higher in rice bran when compared to polished rice. In a similar study, Ye et al. [83] identified ferulic acid, methyl ferulate, *p*-coumaric acid, 5-5′, 8-5′, and 8-*O*-4′ diferulic acid, and 5-5′/8-*O*-4′′dehydrotriferulic acid in the IBPs of brown rice. In addition, the major bound phenolic compounds in sorghum grains and its processing by-products (sorghum bran and sorghum spent grain) were ferulic, *p*-hydroxybenzoic, syringic, caffeic, vanillic, coumaric, and cinnamic acids as well as luteolinidin, apigeninidin, and catechin [84,85]. Furthermore, the pericarps of corn (*Zea mays* L.) genotypes contained 74-83% of IBPs, mainly ferulic, vanillic, isoferulic, syringic, and *p*-hydroxybenzoic acids [17].

Beans and lentils are also valuable sources of IBPs, mainly flavonoids, and their content varied between 41 and 88% of TPC. For instance, numerous bound flavonoids such as catechin, epicatechin, luteolin 3′-7-diglucoside, and catechin-3-glucoside were detected in six lentil cultivars [62]. Similarly, catechin, quercetin glucoside, epicatechin, gallocatechin, syringic acid, and protocatechuic acid were found in the IBPs of red and green lentils and their hulls [12,30,64]. Moreover, green and black lentil hulls contained 50.43–63.11% of IBPs, predominated by myricetin, catechin, quercetin, quercetin glucoside, gallic acid, and protocatechuic acid [44]. However, the IBPs in red and green lentil processing by-products were mainly phenolic acids, including dimethoxybenzoic acid derivative, coumaric acid derivative, gallic acid, and *p*-coumaric acid, and flavonoids, mostly catechin [63]. On the other hand, free, soluble conjugate, and IBPs of 14 beans (black bean, cow gram, chickpea, flower waist bean, kidney bean, mung bean, hyacinth bean, pearl bean, red bean, red kidney bean, *Phaseolus calcaratus*, soybean, semen dolichoris, and spring bay bean) were investigated, and the content of the IBPs varied between 41.17 and 52.59%. In particular, black bean contained isoquercitrin, catechin, quercitrin, protocatechuic acid, *p*-coumaric acid, and vanillic acid [45]. In another study, de Camargo et al. [43] stated that the IBPs of chickpeas made a significant contribution (76.47–87.75%) to the TPC and the major compounds were biochanin A, 3-hydroxybenzoic acid, and taxifolin.

Chia (*Salvia hispanica*) seeds were found to contain a significant amount (32.27%) of IBPs, mainly apigenin (~27%) along with genistein, quercetin-hexoside, *trans*-caffeic acid, *trans*-ferulic acid, and *cis*-hydroxycaffeic acid [47]. Similarly, Mitrović et al. [86] found that the IBPs of chia seeds contributed 27% of TPC; caffeic acid and apigenin 4′-*O*-glucoside were the most dominant phenolic compounds present. Rahman et al. [46] suggested that phenolics in camelina (*Camelina sativa*) and sophia (*Descurainia sophia*) seeds occurred primarily in the esterified form, and were mainly *trans*-sinapic acid and quercetin-hexoside. However, the IBPs of camelina were mainly *trans*-sinapic acid, protocatechuic acid, *p*-hydroxybenzoic acid, quercetin-hexoside, and catechin, whereas sophia seeds were dominated by *trans*-sinapic acid, rosmarinic acid, protocatechuic acid, quercetin-hexoside, and rutin. Likewise, gallic acid, 3,4-dihidroxybenzoic acid, coumaric acid, sinapic acid, ferulic acid, and rutin were detected in the IBPs of black (*Brassica nigra*) and white (*Sinapsis alba*) mustard grains [31]. Additionally, Naczk et al. [87] reported that insoluble tannins dominated in rapeseed/canola hulls and contributed up to 96% of the total tannins. Apart from this, the IBPs in virgin *Camellia oleifera* seed oil comprised 16.5–36.7% of TPC, which were mainly gallic acid and kaempferol derivatives [33].

### 7.3. IBPs in Teas, Coffees, Nuts, Seafoods, and Their By-Products

Teas and coffees are good sources of bound phenolics with strong antioxidant activity. Sun et al. [88] isolated 11 IBPs from the residue of *Apocynum venetum* tea, which were epicatechin, quercetin-3-*O*-β-D-glucopyranoside, kaempferol-3-*O*-β-D-glucopyranoside, loliolide, syringaresinol-4-*O*-β-D-glucopyranoside, alloside of benzyl alcohol, apocynoside I, [1-acetyloxy-3-[3,4,5-trihydroxy-6-[[3,4,5-trihydroxy-6-(hydroxymethyl)oxan-2-yl]oxymethyl]oxan-2-yl]oxypropan-2-yl] hexadecanoate, 3-[(6-*O*-hexopyranosylhexopyranosyl)oxy]-2-(palmitoyloxy)propyl (9Z,12Z,15Z)-9,12,15-octadecatrienoate, 3-hexene-l-*O*-β-D-glucoside, and 1-stearoyl-2-palmitoyl-sn-glycerol. Moreover, the percentage of IBPs in tea seed oils collected from different tea districts in China was 22.2–38.3, and the major compounds were phenolic acids, including benzoic, *p*- hydroxyphenylacetic, vanillic, and cinnamic acids [89]. On the other hand, spent coffee, a major by-product of the brewing process, contained about two-fold more IBPs, mainly caffeoylquinic acids, than the free phenolic fraction [90]. In addition, the IBPs of raw cocoa nibs and husk are a good source of protocatechuic acid, catechin, epicatechin, and epigallocatechin, which contributed up to 40% of the antioxidant activity of the tested samples [73]. Nuts have been characterized as being a rich source of phytochemicals, including phenolic compounds. For instance, the quantity of IBPs in the brown skin of the Brazil nut (*Bertholletia excelsa*) were 19- and 86-fold more than whole nut and kernel, respectively [70]. The major IBPs in the brown skin of the Brazil nut were gallic acid, vanillic acid, protocatechuic acid, catechin, gallocatechin, and taxifolin. Wu et al. [72] stated that 21–38% of the phenolics in the walnut kernel was in bound form, which were mainly phenolic acids such as ellagic, gallic, ferulic, and sinapic acids. Similarly, the IBPs of walnut pellicles contained gallic acid, ellagic acid, protocatechuic acid, *p*-hydroxybenzoic acid, and catechin [71].

Seaweeds are a rich source of phenolic compounds which show a wide range of health-promoting effects. For example, *Sargassum polycystum* obtained from the South China Sea contained a significant content (2.74 mg GAE/g) of IBPs, which was mainly released by alkaline hydrolysis rather than the acid hydrolysis [91]. However, phenolics in sea cucumbers mainly occur in the free form, predominantly phenolic acids and flavonoids [26,27,92]. The IBPs of a sea cucumber (*C. frondosa*) body wall and viscera were around 19% compared to that of the free phenolic fraction. The major IBPs found in these body parts were phenolic acids (chlorogenic acid, *p*-coumaric acid, *p*-hydroxybenzoic acid, gallic acid, gallic acid monohydrate, ellagic acid, and protocatechuic acid) and flavonoids (catechin and quercetin) [26,27].

## 8. Extraction of IBPs

Since IBPs are covalently bound to the cell wall matrix via C-C, ester, and ether bonds, hydrolysis is required to extract/liberate simple phenols (Figure 5).

Acid and alkaline hydrolysis are the most common chemical methods along with enzymatic hydrolysis to release of IBPs from cell wall matrices. In terms of acid hydrolysis, 1–5% HCl/H_2_SO_4_ in water/methanol is a very common approach due to its accessibility and simple steps for extraction. After neutralization and filtration, the extracted IBPs can be used directly for further experimentation due to the ability of acids to break mainly glycosidic bonds or even ester bonds without additional extraction. Nevertheless, phenolics can be degraded during the extraction process or storage, as phenolic compounds are unstable at low pH [2,15]. Moreover, due to the high extraction temperature, thermosensitive compounds (e.g., anthocyanins) could be degraded.

The most popular chemical method for the extraction of IBPs is alkaline hydrolysis, where various concentrations of NaOH/KOH are used to conduct the hydrolysis. Generally, alkaline hydrolysis is able to hydrolyze both ether and ester bonds, which can be conducted at room temperature, causing a low rate of loss of phenolics [2]. However, this is a complex process that requires well-defined conditions to ensure a good recovery. For example, it requires further extraction steps using diethyl ether/ethyl acetate to isolate the liberated phenolics from food matrices. Moreover, this process must be conducted in the dark under a nitrogen/argon atmosphere for 1–4 h to prevent oxidation. Sometimes ascorbic acid or metal chelators (e.g., ethylenediaminetetraacetic acid, EDTA) are also added to control oxidation. Furthermore, the suspension is often acidified (pH 2.0–4.0) with HCl (e.g., 6 M) to prevent the formation of quinone from the deprotonation of phenolic hydroxyl groups in a strong alkali environment. Additionally, foods with high proteins may precipitate under strong basic conditions; hence, a neutral or acidic pH is essential [2,10,15]. In conclusion, alkaline hydrolysis breaks the ester bonds with phenolics and solubilizing proteins, while acid hydrolysis disrupts glycosidic bonds and dissolves sugar moieties, but normally leaves the ester bonds intact.

Apart from chemical methods, enzymatic hydrolysis is also an effective means of liberating IBPs. Usually, carbohydrate-hydrolyzing enzymes such as hemicellulose, cellulase, amylase, xylanase, β-glucosidase, and glucanase are applied to break down cellular matrices containing hemicellulose, cellulose, pectin, and glucan. The main advantage of this technique is the minimum loss of phenolics due to strong acidic/basic conditions during the extraction process [2,4,10]. Moreover, this is a time-saving method along with the higher extraction efficiency of IBPs, though a high cost is always linked with the enzymatic procedures. Aside from this, physical methods (e.g., ultrasounds, microwave, and HPP) coupled with/without chemical/enzymatic procedures have recently been found to be quite effective in releasing IBPs from various food sources, which has been discussed in Section 6.

## 9. Biological Activities of IBPs

Insoluble-bound phenolics have been extensively examined for their biological activities, such as anticancer, antioxidant, anti-inflammatory, angiotensin-converting enzyme (ACE) inhibitory, α-glucosidase, α-amylase, and pancreatic lipase inhibitory, anti-tyrosinase, antiglycation, DNA and LDL oxidation inhibitory, antihyperglycemic, and antimicrobial properties, in cell line experiments as well as in vitro and in vivo models.

### 9.1. Antioxidant Properties

Extensive research has been carried out on the evaluation of antioxidant activities of IBPs using various assays (Table 4).

Most of the IBPs obtained from various sources exhibited strong radical scavenging activities. For instance, ester-linked hydroxycinnamic acid derivatives in oligosaccharides, including coumaric acid, ferulic acid, and syringic acid, have been reported to show radical scavenging effects [93]. Moreover, the IBPs of berry seed meals showed significantly higher antioxidant activity, which was measured using hydroxyl radical scavenging activity, ORAC, reducing power, and metal chelation capacity, compared to free and esterified fractions [53]. Similarly, Gulsunoglu et al. [77] suggested that IBPs extracted from chestnut shells showed higher antioxidant activity than the other fractions. On the other hand, caffeic acid, thomasidioic acid, methyl caffeate, and the 8–5′ DC dehydrodiferulic acid dimer obtained for the IBPs of whole grain brown rice exhibited strong peroxyl radical scavenging capacity, while caffeic acid, methyl caffeate, and the 8–5′ DC dehydrodiferulic acid dimer showed moderate cellular antioxidant activity [82]. The IBPs obtained from barley varieties exhibited a significantly higher antioxidant activity than the other phenolic fractions [41]. Furthermore, IBPs extracted from finger millets were mainly flavonoids, which showed ferric ion reduction and DPPH radical scavenging activities [81]. Lou et al. [65] reported that the antioxidant activity of IBPs in hawthorn berry fruit peels was significantly higher compared to pulps. Apart from this, Yeo and Shahidi [32] found an incremental increase in IBPs in TPC, TFC, DPPH, and ABTS radical cation scavenging activity during 4 days of the germination of lentils. Similarly, the germination of mustard grains improved IBPs and their antioxidant activities, which were measured using FRAP (29%), DPPH (3%), ABTS (160%), and ORAC (42%) assays [31]. The effect of fermentation on the antioxidant activity of pancakes was evaluated and found that the FRAP values of IBPs increased by 30–40% [29]. Mudenuti et al. [73] suggested that the IBPs of cocoa nibs and husk were mainly protocatechuic acid with a 40% contribution to the reducing power. Moreover, sorghum grains, bran, and spent grain contained a higher content of IBPs, mainly phenolic acids linked with non-starch polysaccharides, which showed significant antioxidant activity [84]. The major components of IBPs obtained from red and green lentil processing by-products were phenolic acids with strong antioxidant activities that were measured using FRAP, ORAC, and DPPH assays [88]. In addition, the IBPs of red perilla leaves were mainly caffeic acid, which showed significantly higher antioxidant activity (FRAP and DPPH) than the other phenolic fractions [50]. Suo et al. [69] investigated the antioxidant activity from 19 potato genotypes, and the major IBPs were benzoic and caftaric acids, which showed strong antioxidant activity (FRAP, 10.2-47.0%). In addition, the antioxidant activity of IBPs obtained from tea seed oil accounted for 11.0–49.7, 9.3–38.3, 25.4–43.4, and 10.0–42.9% of the total ABTS, FRAP, DPPH, and ORAC values, respectively [89]. In addition, IBPs also exhibited strong antioxidant activities via hydrogen atom transfer (HAT) and single electron transfer (SET) mechanisms, even achieving around 50% of the total antioxidant capacity of seed oil. Additionally, Zhang et al. [94] reported that the IBPs of rice brans were released at a lower percentage during gastrointestinal digestion (2.68%) than colonic fermentation (27.57%). The released IBPs revealed strong radical scavenging potential, which was examined using DPPH and ABTS assays. On the other hand, hydroxybenzoic acid and protocatechuic acid isolated from the IBPs of barley varieties were the main contributors to antioxidant activities (DPPH, ABTS, and FRAP) [42]. The IBPs of chickpeas decreased oxidative damage in human hepatoma HuH-7 cells by reducing power and inducing peroxyl radicals, suggesting a hepatoprotective potential [43]. Furthermore, the IBPs, mainly gallic acid, of mango leaves inhibited ROS production and cell apoptosis [25].

### 9.2. DNA Oxidation Inhibition

The effects of IBPs on DNA and LDL oxidation as well as α-glucosidase and pancreatic lipase inhibitory activities are summarized in Table 5.

DNA oxidation by free radicals could cause cell mutation and carcinogenesis in humans. Thus, natural antioxidants, mainly phenolic compounds, have been tested for their inhibitory activities against hydroxyl- and peroxyl-induced supercoiled DNA strand scission. For example, the IBPs of the pomegranate divider membrane exhibited the highest inhibition compared to other phenolic fractions against hydroxyl- and peroxyl- radical-induced DNA strand scission with IC_50_ values of 0.06 and 0.05 mg/mL, respectively [67]. This could be due to the scavenging and metal chelation capacities of phenolics against free radicals. Alshikh et al. [62] suggested that the IBPs of lentils showed inhibitory activities against hydroxyl and peroxyl radicals, and this could be related to the presence of *p*-coumaric acid, epicatechin, catechin, and procyanidin dimer B in IBPs. Similarly, IBPs extracted from wood extract, seedling date palm wood, old date palm wood, oak wood, quibracho wood, pinewood, and banana wood showed DNA oxidation inhibitory activity [55]. Moreover, berry, chia, sophia, and camelina seed meals are good sources of IBPs, which have been reported to show inhibitory activity against hydroxyl- and peroxyl-radical-induced DNA oxidation [37,47,53,96]. A strong correlation also existed between TPCs and peroxyl-radical-induced DNA strand scission inhibition. Likewise, the major phenolics in wheat and barley are IBPs, showing strong inhibition against DNA oxidation [37,40]. On the other hand, the IBPs of finger millet showed susceptibility in the breast cancer cell lines MCF-7 and MDA-MB-468, which could be related to the inhibition of DNA oxidation [81]. In addition, sea cucumbers and their processing discards exhibited DNA oxidation inhibition [26,27,59].

### 9.3. LDL Oxidation Inhibition

LDL oxidation is believed to be one of the main reasons for cardiovascular disease (CVD) development. This oxidation could be initiated by the action of metal ions or ROS. Thus, copper-induced LDL oxidation was determined by monitoring the formation of conjugated dienes (CD), which are primary oxidation products. For instance, the effect of IBPs obtained from pomegranate by-products on cupric-ion-induced human LDL oxidation was measured by monitoring the formation of CD and potential antioxidant effects were found [67]. In addition, the IBPs of blackberry showed the highest LDL oxidation inhibition effect (26–60%) compared with other phenolic fractions, which could be due to their free radical scavenging or metal ion chelation properties [53]. Alshikh et al. [62] stated that the IBPs extracted from lentils could inhibit the development of human LDL oxidation up to 67.3% at a very low concentration (0.003 mg/mL). They also found a strong correlation between catechin, epicatechin, procyanidin dimer B-type, or *p*-coumaric acid released from their IBPs and LDL oxidation inhibition. Likewise, tannic acid, syringic acid, gallic acid, and catechin released from the IBPs of wood extract, seedling date palm wood, old date palm wood, oak wood, quibracho wood, pinewood, and banana wood demonstrated inhibitory activity against LDL oxidation [55]. Furthermore, the IBPs of whole grain and bran of wheat completely inhibited (100%) copper-induced LDL oxidation [37]. In addition, insoluble-bound fractions of sea cucumbers and their discards exhibited LDL oxidation inhibition and exhibited a positive correlation between DPPH radical scavenging and metal chelation activities [26,27,59].

### 9.4. α-Glucosidase, α-Amylase, Pancreatic Lipase, and ACE Inhibitory Activities

IBPs have been investigated for their positive effects on the prevention and treatment of metabolic disorders, mainly obesity and diabetes. IBPs have shown inhibitory activities against enzymes related to hyperglycemic effect (e.g., α-glucosidase and α-amylase), high blood pressure (e.g., ACE), and lipolysis in the alimentary system (e.g., pancreatic lipase). For example, Wang et al. [10] summarized the anti-obesity and anti-diabetic effects of IBPs from different sources and found that IBPs showed strong inhibitory effects against α-glucosidase, α-amylase, pancreatic lipase, and ACE enzymes. In particular, α-glucosidase hydrolyzes the 1,4-linked α-D-glucose residues of disaccharide units, thus enabling gastrointestinal absorption. In contrast, pancreatic lipase breaks down triacylglycerols (TAG) into free fatty acids (FFA) and monoacylglycerols (MAG) which could cause the deposition of body fat. Ambigaipalan et al. [67] reported that the IBPs of pomegranate by-products could delay the activity of α-glucosidase and lipase by forming complexes with proteins through hydrogen bonds. Furthermore, da Costa Pina et al. [97] suggested that the IBPs of guarana (*Paullinia cupana*) inhibited the activity of alpha-glucosidase, which was 5.8-fold higher than the soluble counterpart. Additionally, *p*-coumaric, gallic, and ferulic acids extracted from the IBPs of grapefruit peel showed strong α-glucosidase inhibitory activity [34]. Rahman et al. [47] claimed that the α-glucosidase inhibition of chia seed obtained from IBPs was not totally dependent on the quantity of phenolics but rather the certain type of phenolic compound, mainly flavonoids and tannins, and the number of hydroxyl groups in their structures. Likewise, the IBPs of buckwheat, mainly rutin and quercetin, showed strong α-glucosidase inhibition [35]. In addition, Ye et al. [83] suggested that the IBPs of brown rice, mainly 5-5′/8-*O*-4′′ dehydrotriferulic acid and 5-5′ diferulic acid, exhibited strong α-glucosidase inhibitory activities, which could be due to the binding properties of these compounds with the enzyme via hydrogen bonds, ionic bonds, and hydrophobic forces. On the other hand, the inhibitory activities of α-glucosidase and α-amylase by IBPs released from rice bran dietary fiber during gastrointestinal digestion were investigated, and it was found that the IBPs, including ferulic and *p*-coumaric acids, could delay carbohydrate digestion [94]. Furthermore, the IBPs of *Rhus chinensis* fruits were mainly myricitrin and quercitrin, as well as caffeic, caffeoylquinic, ferulic, and *p*-coumaric acid, where the first two compounds were more efficient than others in showing pancreatic lipase inhibitory activity [68]. Apart from this, Wang et al. [10] reported that the IBPs of green pickled olive, red sweet pepper, soybean, white butterfly leaves, pummelo, *Citrus maxima, Veronica persica*, and locust bean exhibited strong ACE inhibitory activity.

### 9.5. Anticancer Effect

IBPs demonstrate anticancer activity by inducing phase II enzymes, mainly NADPH-dependent quinone reductase (QR), working in vivo to inhibit free radicals and electrophiles. For instance, the IBPs of whole wheat can substantially decrease the number of Caco-2 cells without obvious cell death, suggesting antiproliferation functions against colon cancer cells [10]. Moreover, Kuruburu et al. [81] reported the potential of IBPs against breast cancer cell lines obtained from finger millet. Likewise, the IBPs of foxtail millet bran showed inhibitory activity against the growth of human colorectal cancer HCT-116 cells by inducing apoptosis [98]. This could be related to the inhibition of ROS, the blockage of the NF-κB signaling pathway, and the activation of the mitochondria-mediated intrinsic pathway. Furthermore, the IBPs of mango leaves were the most potent antioxidative and cytoprotective components, which were mainly 4-*O*-methylgallic acid, iriflophenone glucoside, mangiferin, *p*-coumaric acid, and catechin gallate [25]. Zhou et al. [24] suggested that the IBPs, mainly caffeic acid, of oil palm fruits showed the potential inhibition of H_2_O_2_-induced ROS generation in HepG2 cells. Likewise, the IBPs of Chinese jujube pulp exhibited antihepatoma effects using HepG2 cells and a nude mice tumor model [99]. Similarly, the IBPs of adlay seed, mostly ferulic acid, could ameliorate H_2_O_2_-induced oxidative stress in HepG2 cells via Nrf2 pathway [100].

### 9.6. Other Effects

IBPs have been examined for their tyrosinase enzyme inhibitory activity. For example, Yao et al. [54] found that the IBPs of raspberry pomace showed anti-tyrosinase activity, where the major IBPs were ellagic, ferulic, *p*-coumaric, and gallic acids. Similarly, the IBPs of sea cucumber tentacles and viscera exhibited anti-tyrosinase activity, which could be related to the copper chelating and free radical scavenging activities of phenolics [27,59]. On the other hand, the IBPs of rice bran dietary fiber significantly reduced the fasting blood glucose and increased glycogen levels in *db/db* mice via activating the insulin signaling pathway [101]. In addition, IBPs were also examined for their antiglycation activity via the monitoring of advanced glycation end products (AGEs). For example, the IBPs of sea cucumbers and their by-products have been reported to have inhibitory activity against the formation AGEs [26,27,59]. This could be due to their ability to chelate metal ions, scavenge free radicals, and trap reactive carbonyl species through adduct formation. In particular, protocatechuic acid, vanillic acid, and quercetin played the main role in showing activity [59].

## 10. Metabolism of IBPs

Phenolic compounds undergo a series of enzymatic reactions and change the physical and chemical properties in the mouth, stomach, small intestine, and large intestine (colon) after consumption. Various enzymes and pH conditions release phenolics from the food matrix in the gastrointestinal tract. The partially/marginally released phenolics penetrate into the intestinal epithelium and reach the blood, depending on their bioaccessibility [102]. The released soluble phenolics could be absorbed in the small intestine (5–10%) and exert bioactivity at the target cell and tissue, while the rest of the free phenolics (90–95%) continue to pass down to the colon along with IBPs and other unabsorbed residues [2]. These phenolics may be metabolized and released via fermentation with the help of gut microbiota. For example, Kroon et al. [103] found that over 95% of the total released ferulic acids in wheat were released during colonic fermentation, while only 2.6% of ferulic acids could be released by gastric and small intestinal digestion. Therefore, IBPs cannot be absorbed in the small intestine as they are attached to insoluble macromolecules, including hemicellulose, cellulose, structural protein, and pectin. IBPs could reach the large intestine directly and undergo fermentation by gut microbiota in order to be released from their bound forms. In particular, microorganisms such as *Lactobacillus* spp. and *Bifidobacterium* spp. release a series of extracellular enzymes, including proteases, carbohydrases, and other types of enzymes, to break covalent bond between phenolics and macromolecules or break down the cellular matrices, followed by liberation of simple phenolics [2]. For instance, Andreasen et al. [104] suggested that the release of insoluble-bond hydroxycinnamic acid could be started by hydrolysis of covalent bonds with the help of esterases in the small intestine. Phenolic compounds found in the colon could inhibit the growth and proliferation of cancer-inducing microorganisms by reducing the colonic pH. For example, the phenolics of blueberries, mainly flavonol and tannin, inhibited the growth of colon cancer cell lines (Caco-2 and HT-29) by approximately 50% [105]. A wide range of products are formed during the fermentation, but only a few of them can be absorbed. However, the absorption of phenolics could occur via passive diffusion and active transport by transporters, though the detailed pathway has not yet been well established. The bioavailability of IBPs is mostly very low, which is mainly linked to the structure of the food matrix which severely delays the hydrolysis of IBPs, causing low bioaccessibility. For example, the urinary recovery content of ferulic acid after wheat bran consumption was found to be as low as 3.1% in humans and 3.9% in rats [4].

## 11. Conclusions

To date, soluble phenolics have been extensively studied, while a large proportion of the IBPs found in food matrices are neglected. In particular, plenty of IBPs are found in the protective tissues of foods such as hulls, husks, and brans, and are removed during processing (e.g., milling and polishing), resulting in the loss of valuable biomolecules. These IBPs, such as phenolic acids and flavonoids, exhibit strong bioactivities, including antioxidant, anti-inflammation, anticancer, and cardiovascular disease ameliorating effects. However, current approaches available to liberate and assess IBPs are often destructive and ineffective. Hence, novel techniques should be further developed in order to extract IBPs without changing their contents, functions, and organoleptic properties. Moreover, IBPs are hardly digested in the human digestive tract; hence, the interaction between IBPs and gut microbiota and their bioavailability, metabolism, and mechanism of action need to be critically examined.

## Figures and Tables

**Figure 1 antioxidants-12-00203-f001:**
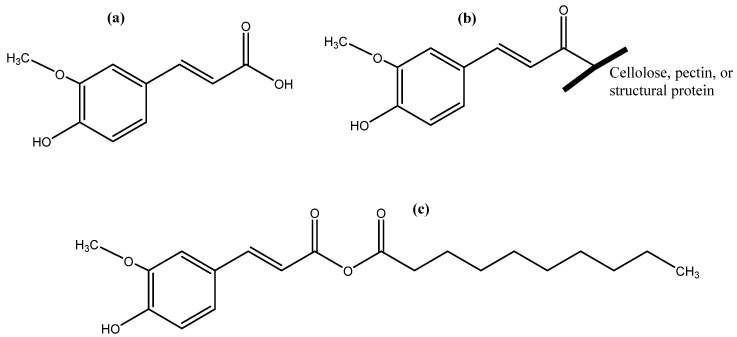
Free (**a**), bound (**b**), and ester form (**c**) of ferulic acid.

**Figure 2 antioxidants-12-00203-f002:**
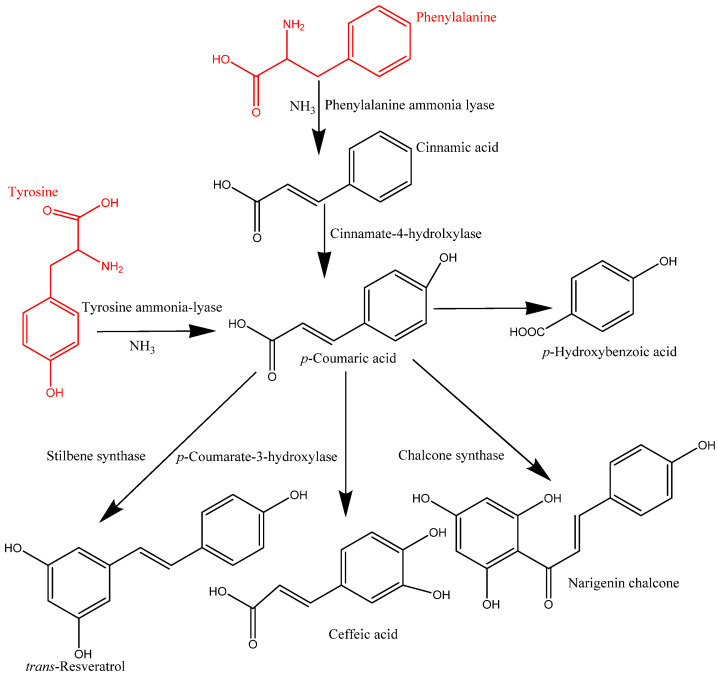
Synthesis of phenolic compounds.

**Figure 3 antioxidants-12-00203-f003:**
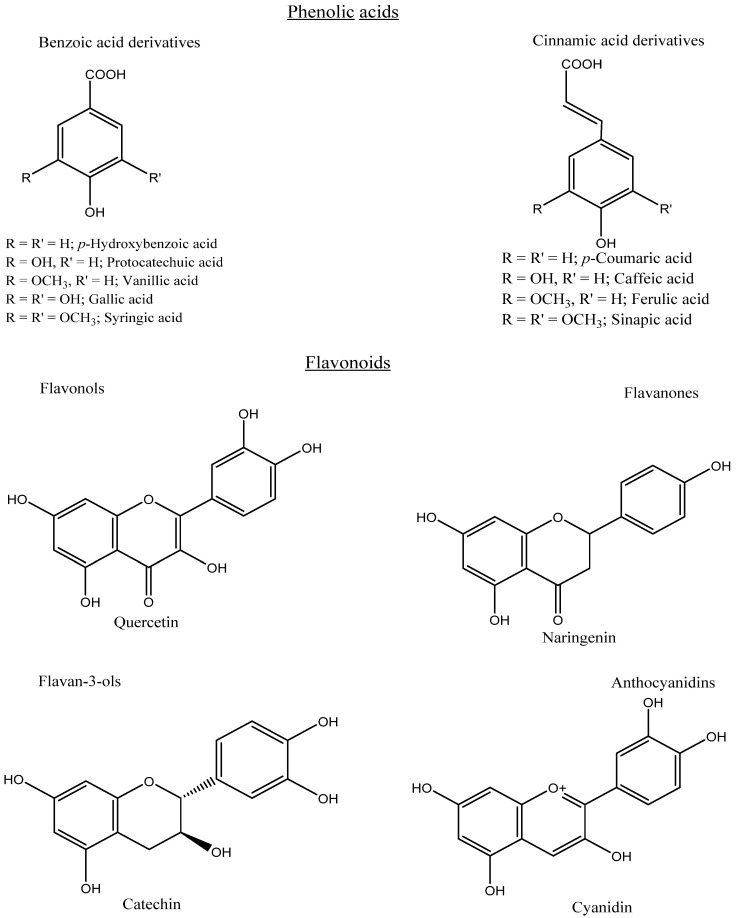
Chemical structures of some phenolic acids and flavonoids.

**Figure 4 antioxidants-12-00203-f004:**
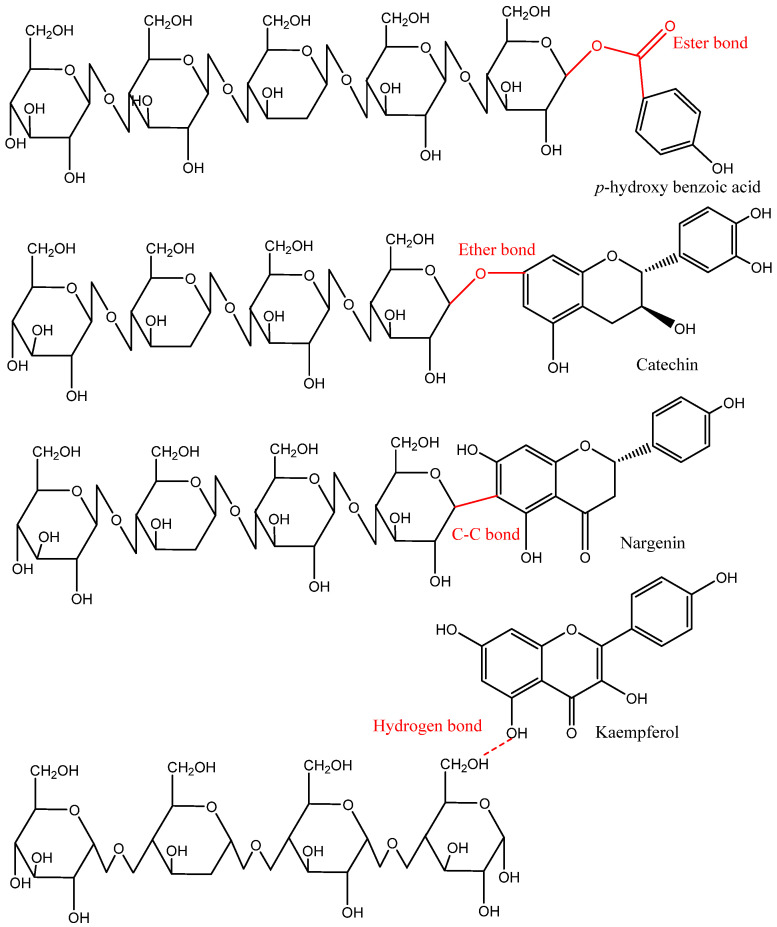
Representative covalent and hydrogen bonds found in the insoluble-bound phenolics and food matrix [2].

**Figure 5 antioxidants-12-00203-f005:**
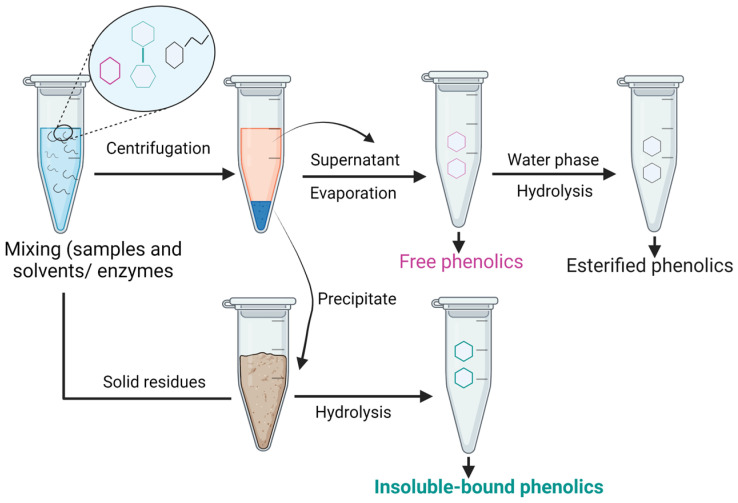
Schematic diagram of IBPs extraction.

**Table 1 antioxidants-12-00203-t001:** Release of insoluble-bound phenolics from cell wall matrix upon processing.

Sources	Processing Techniques	Content of IBPs	Antioxidant Activity of IBPs	Phenolic Profiles in IBPs	References
Oil palm (*Elaeis guineensis* Jacq.) fruits	Ultra-high pressure (UHP, 500 MPa for 10 min)	Increased TPC and TFC by around 2 times upon UHP	Increased DPPH and ABTS radical cation scavenging activities, FRAP values, and ROS inhibitory activity	Increased the content and number of individual phenolics	[24]
Mango leaves	UHP (500 MPa for 10 min)	Increased TPC and TFC significantly upon UHP	Increased DPPH and ABTS radical cation scavenging activities and FRAP values	Increased the content and number of individual phenolics	[25]
Sea cucumber (*C. frondosa*) body wall	High-pressure processing (HPP, 200, 400, and 600 MPa for 5, 10, and 15 min)	Increased IBPs in TPC and TFC by about 27 and 35%, respectively	Increased DPPH radical scavenging and metal chelation activities	Increased the content (~28%) and number of individual phenolics	[26]
Sea cucumber (*C. frondosa*) waste	HPP (600 MPa for 10 min)	Increased the overall TPC and TFC	Increased DPPH radical scavenging activity but decreased ABTS radical cation scavenging activity	Increased the content (~26%) and number of individual phenolics	[27]
Whole grain rice (black, red, and white)	Gamma (γ)-irradiation (10 kGy)	Bound phenolics increased significantly compared to free phenolic fraction	Increased ABTS radical cation scavenging activity	NA	[28]
Fermented pancake (*Injera*)	Fermentation	Increased TPC and decreased TFC	Increased FRAP values but decreased DPPH and ABTS radical cation scavenging activities	Decreased the content of individual phenolics by 2–100%	[29]
Lentil hulls	Fermentation	IBPs decreased significantly	NA	Individual phenolic compounds decreased upon fermentation	[30]
Mustard grains (*Brassica nigra* and *Sinapsis alba*)	Germination	The TPC and TFC increased or remained the same	Increased DPPH and ABTS radical cation scavenging activities and FRAP and ORAC values for *S. alba*	Showed an overall positive effect on the phenolic profile	[31]
Lentils	Germination	The TPC and TFC increased upon processing	Increased DPPH and ABTS radical cation scavenging activities	NA	[32]
Virgin (*Camellia oleifera)* seed oil	Thermal pre-treatment (0–120 min for 90 and 150 °C)	Fluctuated among different heating conditions	NA	Fluctuated among different heating conditions	[33]
Hawthorn fruit	Thermal processing: lightly cooked (80 °C for 20 min and 100 °C for 15 min) and well-cooked (120 °C for 20 min and 150 °C for 15 min)	IBPs increased by 55.84 and 30.35% through being lightly and well-cooked, but overall TPC decreased with cooking	Increased ORAC values but decreased DPPH and ABTS radical cation scavenging activities by both treatments	Decreased the number of individual phenolics and increased the content only by lightly cooking	[11]
Lentils	Hydrothermal processing (boiling for 25 min)	Decreased TPC and TFC	Decreased ORAC values, DPPH radical scavenging activity, and reducing power ability	Decreased the overall content and number of phenolics	[12]
Grapefruit peels	Microwave and enzymatic treatments	Improved the overall TPC and TFC	Improved DPPH radical scavenging activity and ORAC values	Combined microwave and enzymatic treatment improved the release of phenolic acids	[34]

NA, not available; ROS, reactive oxygen species; TPC, total phenolic content; TFC, total flavonoid content; DPPH, 2,2-diphenyl-1-picrylhydrazyl; ABTS, 2,2′-azinobis(3-ethylbenzothiazoline-6-sulfonic acid); FRAP, ferric reducing antioxidant power; and ORAC, oxygen radical absorbance capacity.

**Table 2 antioxidants-12-00203-t002:** Distribution of IBPs in various food matrices.

Sources	Free (mg GAE/g)	IBPs (mg GAE/g)	TPC (mg GAE/g)	Ration (IBPs/F)	IBPs/TPC (%)	References
Buckwheat	5.18–13.74	0.63–0.96	6.29–14.4	0.07–0.12	6.67–10.01	[35]
Buckwheat brans	1242.49 (mg/kg)	689.89 (mg/kg)	1932.3 (mg/kg)	0.56	35.7	[36]
Wheat brans (soft and hard)	0.84–0.98 (mg FAE/g)	11.3–12.18 (mg FAE/g)	13.51–14.59 (mg FAE/g)	13.45–12.42	83.48–83.64	[37]
Millets	0.007–0.032 (mmol FAE/g)	0.002–0.081 (mmol FAE/g)	0.009–0.11 (mmol FAE/g)	0.28–2.53	22.23–73.63	[38]
Millet Seeds	0.004–0.025 (mmol FAE/g)	0.001–0.062 (mmol FAE/g)	0.005–0.087 (mmol FAE/g)	0.25–2.24	20–71.26	[39]
Barley varieties	0.18–0.42 (mg FAE/g)	2.03–3.36 (mg FAE/g)	2.63–4.51 (mg FAE/g)	8–11.27	74.50–77.18	[40]
Barley varieties	0.037–0.16	0.21–0.30	0.28–0.52	1.87–5.67	57.69–75	[41]
Barley varieties	1.66–2.37	1.70–2.40	3.36–4.53	1.01–1.02	49.40–52.31	[42]
Corn varieties (pericarp)	0.013–-0.021 (mmol FAE/g)	0.27–0.43 (mmol FAE/g)	0.28–0.45 (mmol FAE/g)	20.47–20.76	95.56–96.42	[17]
Chickpeas	0.073–3.28	0.13–17.98	0.17–20.49	1.78–5.48	76.47–87.75	[43]
Lentil hulls (green and black)	31.49–40.26	40.96–53.88	81.22–85.37	1.3–1.33	50.43–63.11	[44]
Lentil hulls (raw)	3.22–4.03	3–3.64	6.22–7.68	0.86–0.93	47.39–48.23	[12]
Lentils	3.13–4.25	4.78–6.45	8.13–10.69	1.39–1.66	58.79–60.33	[32]
Beans	0.14–0.52	0.14–0.81	0.34–1.54	1–1.55	41.17–52.59	[45]
Camelina (*Camelina sativa*)	4.07	0.82	11.69	0.2	7.01	[46]
Sophia (*Descurainia sophia*)	4.14	2.5	22.4	0.6	11.16	[46]
Chia (*Salvia hispanica*) seeds	8.69	4.59	14.22	0.52	32.27	[47]
Flowers (*Lonicera japonica* and *L. macranthoides*)	0.15 mmol GAE/g	0.006 mmol GAE/g	0.19 mmol GAE/g	0.04	3.15	[48]
Flowers (*Camellia oleifera* and *C. polyodonta*)	102.68–137.9	1.19–2.04	104.72–138.96	0.01	1.13–1.46	[49]
Leaves (*Lonicera japonica* and *L. macranthoides*)	0.098 mmol GAE/g	0.029 mmol GAE/g	0.15 mmol GAE/g	0.29	19.92	[48]
Leaves (green perilla)	34.18	5.08	45.03	0.14	11.28	[50]
Leaves (red perilla)	12.38	17.8	35.44	0.5	49.04	[50]
Fruit leaves (*Averrhoa carambola*)	6.27	16.11	29.96	2.53	53.77	[51]
Fruit leaves (*Artocarpus heterophyllus*)	2.76	20.81	28.67	7.53	72.58	[51]
Stem and root (*Terminalia sericea*)	15.12	10.38–11.62	25.5–26.74	0.68–0.76	40.70–43.44	[52]
Berry seeds (blackberry)	2.23	7.93	13.6	3.55	58.3	[53]
Berry seeds (black raspberry)	0.8	4.6	7.3	5.75	63.01	[53]
Berry seeds (raspberry)	8.84	7.31	25.4	0.82	28.77	[54]
Pomace (raspberry)	8.66	6.39	24.14	0.73	26.47	[54]
Wood (seedling date palm)	80.03	21.05	101.08	0.26	20.82	[55]
Fruit (*Pyrus pashia* Buch) pulp (Kainth)	1.78	7.07	10.36	3.97	68.24	[56]
Fruit (*Annona crassiflora*) peel (araticum)	1.79	6.31	31.65	3.52	19.93	[57]
Fruit (*Annona crassiflora*) pulp (araticum)	1.41	9.04	20.49	6.41	44.11	[57]
Mistletoes (*Viscum articulatum* and *V. liquidambaricolum*)	0.008–0.009 mmol FAE/g	0.003–0.004 mmol FAE/g	0.012–0.014 mmol FAE/g	0.37–9.44	25–28	[58]
Dried hawthorn (*Crataegus pinnatifida*)	29.34	0.47	29.81	0.016	1.57	[11]
Sea cucumber (*Cucumaria frondosa*) body wall	2.2	0.74	3.98	0.33	18.59	[26]
Sea cucumber (*C. frondosa*) viscera	2.27	0.56	3.02	0.24	18.54	[27]
Sea cucumber (*C. frondosa*) tentacles	2.41	0.38	3.09	0.15	12.29	[59]

GAE, gallic acid equivalents and FAE, ferulic acid equivalents.

**Table 3 antioxidants-12-00203-t003:** Individual phenolic compounds in various food matrices.

Sources	Total Bound Phenolics (µg/g)	Major Bound Phenolics (µg/g)	References
Buckwheat brans	689.81	Catechin (207.74), syringic acid (85.86), epicatechin (59.08), rutin (51.64), swertiamacroside (39.40), and quercitrin (26.64)	[36]
Buckwheat	NA	Rutin (85.02–416.83), dihydromyricetin (57.85–299.93), kaempferol-3-*O*-rutinoside (43.34–230.85), *p*-hydroxybenzoic acid (61.57–193.72), gallic acid (59.79–71.78), and syringic acid (4.28–66.97)	[35]
Purple wheat (*Triticum aestivum*) fine brans	390	*trans*-Ferulic acid (279), *cis*-ferulic acid (25.6), *trans*-*p*-coumaric acid (9.24), and sinapic acid (7.76)	[60]
Millet seeds	NA	Ferulic acid (132.1–1290) and *p*-coumaric acid (14.9–778.5)	[39]
Millets	NA	Ferulic acid (178.82–1685.04-1290) and *p*-coumaric acid (20.68–1139.06)	[38]
Grain hulls	NA	Ferulic acid (266.9–744.2), sinapic acid (1.35–15.72), chrysoeriol-7-*O*-glucuronide (10.86–64.93), and luteolin (3.01–12.56)	[61]
Barley varieties	1626.19	Gallic acid (338.29), benzoic acid (285.79), syringic acid (267.47), naringenin (128.83), *p*-coumaric acid (127.92), and hesperidin (102.05)	[42]
Corn (quality protein corn)	8675	Ferulic acid (3522), vanillic acid (2317), isoferulic acid (901), syringic acid (897), and *p*-hydroxybenzoic acid (532)	[17]
Chickpeas	NA	Biochanin A (117.9-841.9), 3-hydroxybenzoic acid (143.2–319.1), and taxifolin (22.9-56.6)	[43]
Lentils		Procyanidin dimer B (35.7–167), catechin (15–78.4), epicatechin (0.5–7.94), and catechin-3-glucoside (15.1–122)	[62]
Lentils (red and green)	1446.80–2204.31	Dimethoxybenzoic acid derivative (630.76–953.95), coumaric acid derivative (103.97–243.96), catechin (11.22–278.77), gallic acid (186.48–230.31), and *p*-coumaric acid (83.17–173.61)	[63]
Lentils	NA	Catechin (320–2170), protocatechuic acid derivative (160–520), and epicatechin (80–290)	[12]
Lentil hulls	NA	Syringic acid (7180–21560), protocatechuic acid (5780–19090), quercetin (5040–14940), catechin (6670–9700), and gallocatechin (5170–7310)	[30]
Lentil hulls	6710–10340	Catechin (3770–9130), protocatechuic acid (1580–1940), quercetin glucoside (590–1100), and epicatechin (270–620)	[64]
Lentils (hull, whole, and dehull)	48.7–2812.1	Myricetin (2.1–653.4), catechin (3.1–534.1), gallic acid (0.9–489.9), protocatechuic acid (4.6–439.1), quercetin (3.20–320.7), and quercetin glucoside (1–250.5)	[44]
Beans (black)	1388.71	Isoquercitrin (462.36), protocatechuic acid (253.42), catechin (109.70), *p*-coumaric acid (108), vanillic acid (100.22), and quercitrin (86.61)	[45]
Camelina (*Camelina sativa*)	316.12	*trans*-Sinapic acid (172.02), quercetin-hexoside (48.49), protocatechuic acid (31.11), *p*-hydroxybenzoic acid (16.60), and catechin (12.49)	[46]
Sophia (*Descurainia sophia*)	187.45	*trans*-Sinapic acid (70.48), rosmarinic acid (31.03), quercetin-hexoside (21.54), rutin (21.54), and protocatechuic acid (17.24)	[46]
Chia seeds	578.29	Apigenin (152.51), genistein (91.98), quercetin-hexoside (91.05), *trans*-caffeic acid (72.02), *trans*-ferulic acid (69.70), and *cis*-hydroxycaffeic acid (67.44)	[47]
Leaves (UHP-treated mango)	NA	Mangiferin (18201.35), iriflophenone glucoside (11915.92), catechin gallate (7203.58), gallic acid (6127.62), isoquercitrin (5874.87), 4-*O*-methylgallic acid (3887.52), homomangiferin (3611.83), quercitrin (2850.16), *p*-coumaric acid (2470.37), and dihydroquercetin (1094.64)	[25]
Leaves (*Mangifera indica*)	12619.9	Epicatechin (7697.95), gallic acid (2424.90), rutin (977.63), and isoquercitrin (605.60)	[51]
Leaves (*Lonicera macranthoides*)	3190	Caffeic acid (1150), luteoloside (1210), and isoquercitrin (930)	[48]
Flowers (*Camellia oleifera* and *C. polyodonta*)	NA	Gallic acid (101.23–580.10), *p*-coumaric acid (274.88–423.32), astragaline (91.26–304.61), kaempferol-3-*O*-rutinoside (59.43–119.08), and quercitrin (7.03–116.77)	[49]
Seed (black raspberry)	NA	Quercetin 3-*O*-glucoronide (11.49), quercetin (3.26), epicatechin (3.11), *p*-coumaric acid (2.43), gallic acid (2.42), caffeic acid (1.58), epigallocatechin (1.36), and protocatechuic acid (0.93)	[53]
Fruit (*Annona crassiflora*) peel (araticum)	1367.6	Catechin (812.36), epicatechin (327.31), and protocatechuic acid (125.06)	[57]
Grapefruit peel (microwave and enzymatic treatment)	NA	Gallic acid (42.50), naringin (21.54), ferulic acid (18.46), and protocatechuic acid (6.16)	[34]
Fruit (*Crataeguspinnatifida*) peel (hawthorn)	1000.34	Epicatechin (265.63), caffeic acid (111.02), catechin (387.23), *p*-coumaric acid (85.82), and protocatechuic acid (36.04)	[65]
Fruit (*Annona crassiflora*) pulp (araticum)	716.23	Catechin (405.54), epicatechin (239.32), and protocatechuic acid (62.89)	[57]
Fruit (*Pyrus pashia* Buch) pulp	NA	Catechin (0.44), epicatechin (0.29), procyanidin B_2_ (0.08), and *p*-coumaric acid (0.02)	[56]
Lychee pulps	NA	Syringate (12.83–67.14), vanillic acid (7.4–66.58), caffeic acid (54.48–66.51), catechin (12.97–19.95), and epicatechin (12.7–18.19)	[66]
Raspberry pomace	1323.96	Gallic acid (604.65), ellagic acid (452.44), ferulic acid (76.67), *p*-coumaric acid (56.67), protocatechuic acid (46.76), and catechin (18.67)	[54]
Fruit (pomegranate) outer skin	28.67	Gallic acid (11.31), kaempferol 3-*O*-glucoside (9,67), brevifolin carboxylic acid (3.34), *trans*-*p*-coumaric acid (1.17), vanillic acid (1.07), and protocatechuic acid (1.06)	[67]
Fruit- hawthorn (*C. pinnatifida*)	66020	Procyanidin B2 (36030), rutin (27120), isoquercetin (13870), chlorogenic acid (9920), and hyperoside (6200)	[11]
Fruit (*Rhus chinensis*)	99560.4	Quercitrin (36098.16), gallic acid (1400.92), and myricitrin (425.33)	[68]
Fruits-oil palm (*Elaeis guineensis*)	NA	Caffeic acid (11269.66), *p*-hydroxybenzoic acid (3605.47), catechin (692.87), ferulic acid (628.79), hesperetin (601.93), *p*-coumaric acid (531.82), epigallocatechin (448.23), protocatechuic acid (365.05), and gallic acid (311.16)	[24]
Mistletoes (*Viscum articulatum* and *V. liquidambaricolum*)	822.3–1135.76	Epigallocatechin (14.63–223.32)*, p*-coumaric acid (14.26–206.97), ferulic acid (97.94–171.18), catechin hydrate (92.21–129.17), *trans*-cinnamic acid (46.03–124.38), kaemferol (18.15–99.4), myricetin (33.14–75.23), quercetin (41.44–62.30), *p*-hydroxybenzoic acid (48.02–55.2), vanillic acid (37.4–52.73), and caffeic acid (28.2–49.88)	[58]
Potatoes	NA	Rutin (36.77–1995.73), benzoic acid (263–1831.84), caftaric acid (21.55–940.77), and cryptochlorogenic acid (4.53–32.39)	[69]
Brazil nut (brown skin)	7873.04	Catechin (2874.55), gallic acid (1638.92), protocatechuic acid (1319.95), gallocatechin (1316.32), taxifolin (333.16), and vanillic acid (285.53)	[70]
Walnut pellicle	NA	Gallic acid (234–1142), ellagic acid (432–509), catechin (40.3–89.1), protocatechuic acid (23.6–81.6), and *p*-hydroxybenzoic acid (25.2–78.3)	[71]
Walnut kernel	NA	Ellagic acid (46.38–93.27), gallic acid (3.78–4.58), ferulic acid (3.18–3.79), and sinapic acid (1.93–2.57)	[72]
Cocoa (nibs and husk)	NA	Protocatechuic acid (5400–12200), catechin (100–1400), epigallocatechin (300–400), and epicatechin (100–200)	[73]
Sea cucumber (*Cucumaria frondosa*) body wall	175	Protocatechuic acid (24), catechin (18), *p*-coumaric acid (17), gallic acid (17), *p*-hydroxybenzoic acid (15), quercetin (15), and ellagic acid (14)	[26]
Sea cucumber (*C. frondosa*) viscera	259.2	Chlorogenic acid (30.6), *p*-coumaric acid (28.8), hydroxygallic acid (24.2), catechin (23.3), ellagic acid (21.3), and protocatechuic acid (20.5)	[27]

**Table 4 antioxidants-12-00203-t004:** Antioxidant potential of IBPs obtained from various sources.

Sources	DPPH RSA (µmol TE/g)	ABTS^+^ RSA (µmol TE/g)	Hydroxyl RSA (µmol TE/g)	Metal Chelation (µmol EDTAE/g)	ORAC (µmol TE/g)	TEAC (µmol TE/g)	FRAP (µmol TE/g)	Reducing Power (µmol TE/g)	References
Buckwheat	4.3–7.68	7.12–11.54	13.23–14.54	NA	NA	NA	NA	NA	[35]
Millet Seeds	2.77–17.38 (µmolFE/g)	NA	49.82–1110.2 (µmol FE/g)	NA	44.2–606.88 (µmol FE/g)	NA	NA	NA	[39]
Millets	NA	NA	NA	NA	NA	6.77–86.13		2.96–29.33 (µmol AAE/g)	[38]
Grain hulls	318.53–607.81 (µg TE/g)	197.3–880.28 (µg TE/g)	NA	NA	NA	NA	NA	NA	[61]
Barley varieties	3.95–5.62	NA	NA	NA	22.13–34.67	7.44–9.88	NA	NA	[40]
Corn (quality protein corn) pericarp	2047	958	NA	NA	NA	NA	43.3		[17]
Wheat brans (soft and hard)	634.6–661.5	NA	NA	NA	10550–11350	28270–32765	NA	NA	[37]
Lentils	40–420	NA	40–300	NA	NA	90–930	NA	20–270	[62]
Lentils (raw)	474–551 (µg TE/g)	NA	NA	NA	1355–2144 (µg TE/g)	NA	NA	765–872 (µg AAE/g)	[12]
Lentil hulls	263–719 (µg TE/g)	NA	NA	174–202 (µg CE/g)	NA	NA	NA	446–5455 (µg AAE/g)	[64]
Lentils (hull, whole, and dehull)	200–5600 (µg TE/g)	20–1060 (µg TE/g)	1620–3550 (µg TE/g)	NA	NA	NA	NA	NA	[44]
Beans (black)	2.38	5.7	NA	NA	NA	NA	NA	NA	[45]
Camelina (*Camelina sativa*)	NA	NA	NA	13.87	NA	14.11	NA	6.44	[46]
Sophia (*Descurainia sophia*)	NA	NA	NA	6.91	NA	39.54	NA	26.05	[46]
Chia seeds	9.98		17.9	2.5	NA	58.35	NA	37.19	[47]
Leaves (*Mangifera indica*)	149.7	365.13	NA	NA	NA	NA	213.88	NA	[51]
Leaves (*Lonicera macranthoides*)	12.02	20.21	NA	NA	248.16	NA	213.88	NA	[48]
Flowers (*Camellia oleifera* and *C. polyodonta*)	39.6–54.57 (µg TE/g)	2144.75–4085.57 (µg TE/g)	NA	NA	NA	NA	7.52–19.50 (µg TE/g)	NA	[49]
Seed (blackberry)	NA	NA	53.8	68.6 (µmol TE/g)	32.8	NA	NA	52.2	[53]
Fruit (*Annona crassiflora*) peel (araticum)	41.37	NA	NA	NA	117.28	63.03	NA	NA	[57]
Grapefruit peel (microwave and enzymatic treatment)	0.23	0.31	NA	NA	4.3	NA	NA	NA	[34]
Fruit (*Annona crassiflora*) pulp (araticum)	55.57	NA	NA	NA	119.19	99.07	NA	NA	[57]
Fruit (*Pyrus pashia* Buch) pulp	13.8 (IC_50_ μg/mL)	12.22 (IC_50_ μg/mL)	NA	NA	NA	2970 (μg TE/g)	2159 (μg TE/g)	NA	[56]
Fruit (pomegranate) outer skin	230	11.72	29.31 (µmol GAE/g)	0.86	48.44	NA	NA	NA	[67]
Hawthorn fruit (*C. pinnatifida*)	27.74 (IC_50_ mg/mL)	7.31 (IC_50_ mg/mL)	NA	NA	NA	NA		NA	[11]
Mistletoes (*Viscum articulatum* and *V. liquidambaricolum*)	1.51–1.83 (µmol FAE/g)	NA	NA	NA	NA	1.4–5.78	8.07–10.31 (µmol FAE/g)	NA	[58]
Wood (seedling date palm)	NA	NA	NA	12.95	NA	NA	NA	810	[55]
Brazil nut (brown skin)	29.13 (µmol CE/g)	NA	101.26 (µmol CE/g)	NA	168.35	59.83	NA	39.9 (µmol AAE/g)	[70]
Walnut kernel	35.25–49.97 (IC_50_ μg/mL)	NA	NA	NA	NA	NA	NA	NA	[72]
Sea cucumber (*Cucumaria frondosa*) body wall	949 (µg TE/g)	1187 (µg TE/g)	2017 (µg TE/g)	92 (µg TE/g)	NA	NA	NA	NA	[26]
Sea cucumber (*C. frondosa*) viscera	727.6 (µg TE/g)	947.6 (µg TE/g)	2543 (µg TE/g)	72.7 (µg TE/g)	NA	NA	NA	NA	[27]

RSA, radical scavenging activity; ORAC, oxygen radical absorbance capacity; TEAC, Trolox equivalent antioxidant capacity; FRAP, ferric reducing antioxidant power, TE, Trolox equivalents; GAE, gallic acid equivalents; EDTAE, ethylenediaminetetraacetic acid equivalents; FAE, ferulic acid equivalents; AAE, ascorbic acid equivalents; and CE, catechin equivalents.

**Table 5 antioxidants-12-00203-t005:** Biological activities of IBPs obtained from various sources.

Sources	DNA Oxidation		LDL Oxidation	α-Glucosidase	Pancreatic Lipase	References
	Peroxyl Radical	Hydroxyl Radical				
Pomegranate outer skin	IR: 48.80%, IC_50_: 0.1 mg/mL	IR: 16.11%, IC_50_: 0.32 mg/mL	NA	IR: 2.50%, IC_50_: 20.66 mg/mL	IR: 0.81%, IC_50_: 64.38 mg/mL	[67]
Pomegranate mesocarp	IR: 18.44%, IC_50_: 0.27 mg/mL	IR: 11.17%, IC_50_: 0.45 mg/mL	NA	IR: 5.08%, IC_50_: 9.86 mg/mL	IR: 4.34%, IC_50_: 11.57 mg/mL	
Pomegranate divider	IR: 98.42%, IC_50_: 0.05 mg/mL	IR: 79.09%, IC_50_: 0.06 mg/mL	NA	IR: 5.95%, IC_50_: 8.45 mg/mL	IR: 15.16%, IC_50_: 3.31 mg/mL	
Lentil cultivars	IR: 69.64–88.01%	NA	NA	NA	NA	[62]
Lentils (black and green)	IR: 7.7–89.8 mg CE/g	IR: 1.62–3.55 mg CE/g	NA	NA	NA	[44]
Date palm wood	IR: 86.39%	IR: 38.64%	NA	NA	NA	[55]
Winemaking by-products	NA	NA	NA	IR: 90–100%	IR: 40-50%	[95]
Berry seed meals	NA	NA	IR: 48.51–59.93%	NA	NA	[53]
Chia seeds	IC_50_: 5.26 mg/mL	IC_50_: 25.74 mg/mL	IC_50_: 0.07 mg/mL	IC_50_: 192.54 mg/mL	IC_50_: 17.10 mg/mL	[47]
Sophia seed meals	IC_50_: 2.42 mg/mL	IC_50_: 15.74 mg/mL	IC_50_: 0.02 mg/mL	IC_50_: 152.8 mg/mL	IC_50_: 12.23 mg/mL	[96]
Camelina seed meals	IC_50_: 5.40 mg/mL	IC_50_: 5.06 mg/mL	IC_50_: 12.23 mg/mL	IC_50_: 128.39 mg/mL	IC_50_: 4.15 mg/mL	
Wheat	NA	IR: 920–1740 µg/g	IR: 6502–25600 µg/g	NA	NA	[37]
Barley	IR: 82.34–96.66%	NA	IR: 42.92–72.32%	NA	NA	[40]
Guarana powder	NA	NA	NA	IC_50_: 1.62 µg GAE/mL	NA	[97]
Sea cucumber viscera	IR: 80.48%	IR: 66.50%	IR: 20.27%	IR: 26.15%	NA	[27]
Sea cucumber tentacles	IR: 80.07%	IR: 68.1%	IR: 15.95%	IR: 26.39%	NA	[59]
Sea cucumber body wall	IR: 85.80%	IR: 72.81%	IR: 34.82%	IR: 34.83%	NA	[26]

IR, inhibitory rate; NA, not available, CE, catechin equivalents; and GAE, gallic acid equivalents.

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
