# Peer review of "Importance of Insoluble-Bound Phenolics to the Antioxidant Potential Is Dictated by Source Material"

_antioxidants, 2023, doi:10.3390/antiox12010203_

Round 1

Reviewer 1 Report

Here are my suggestions for authors.

General comments:

- I would strongly suggest to authors to reorder last section about extraction and to place it before any biological activity of IBPs and after general chemistry. It seems to me much more logical to firstly explain how we can extract these compounds and than to discuss about their biological activity.

- DPPH is a radical while ABTS is a radical cation. Please label them in adequate way through a whole Manuscript to be chemically correct.

All specific comments are listed below with an appropriate Line number(s) from text in order to facilitate tracking.

Specific comments

Line 26: Suggest to delete "food". It is surplus here.

Line 43: I think that abbreviation should be CVDs since diseases are in plural? Just like IBPs. Check/correct.

Line 101: Delete "and" in front of "are responsible". It is surplus here.

Line 122: Suggest to add "rather" in front of "occur" here.

Line 125: It should be "C-2 moiety" here. Replace.

Line 129: I am not sure that can agree about that flavonoids are "water soluble" compounds. It is well known that these compounds are quite insoluble in water in aglycone form. Please check/correct. given statement.

Lines 160-161: Maybe "glycosides" instead of "glucosides" if you are talking about general group of compounds? Glucoside is compound with glucose. Please check/correct.

Line 174: "simple phenols" in plural here. Correct.

Line 188: Please be consistent. It should be always maize or corn.

Line 203: Can authors provide here some data about differences inside the same variety as they said previously?

Lines 220-222: Interactions of phenolics from different plant food stuff with proteins (in particular from different milks) are intensively elaborated in last couple years in the literature. In that way I would like to ask authors to expand here literature and include additional the most novel data about this.

Line 238: Please define all abbreviations used in the Table 1. It should be self-explanatory unrelated to the rest of the text.

Line 340: In the Table 3 correct typos- some names (not at the beginning of text) are written with capital letter which should not be. For instance, for purple wheat (ref. no. 60) it should be "trans-ferulic acid" without capital letter F. The same for reference no. 46, etc.

Line 341: Please define abbreviations DRSA, and DPPH here just like you did for the rest of abbreviations.

Line 347: typo- put Latin name in Italic here.

Line 435: The same as previous.

Line 443: "Triticum aestivum L." with point after L. Correct.

Lines 609-610: Please check verb. I think it should be "is summarized" in the Line 610 since you are referring to "The effect" in the Line 609? Or "The effects .... are summarized". Please check/correct.

Kind regards.

Reviewer 2 Report

The review deals with a recent and extensively studied topic, as witnessed by the authors' recent bibliography. The manuscript chemically and biologically analyses IBPs in food matrices. I believe this review is ready for publication.

Reviewer 3 Report

This is a very interesting manuscript and should be published. The authors have done a review that highlights a very important issue, especially in the context of the biological properties of food. In principle, I have no major comments on this manuscript. However, it seems to me that figure 5 should be better described (I understand that the supernatant is evaporated and the precipitate is hydrolyzed). It is logical but the figure could be clearer.
